# ATOMNAS: FINE-GRAINED END-TO-END NEURAL ARCHITECTURE SEARCH

**Jieru Mei**[1*]**, Yingwei Li**[1*]**, Xiaochen Lian**[2]**, Xiaojie Jin**[2]**, Linjie Yang**[2]**,
Alan Yuille**[1] **& Jianchao Yang**[2]
[1]Johns Hopkins University
[2]ByteDance AI Lab

`meijieru@gmail.com, yingwei.li@jhu.edu, {xiaochen.lian, jinxiaojie, linjie.yang}@bytedance.com,`
`alan.l.yuille@gmail.com, yangjianchao@bytedance.com`

## ABSTRACT

Search space design is very critical to neural architecture search (NAS) algorithms. We propose a fine-grained search space comprised of atomic blocks, a minimal search unit that is much smaller than the ones used in recent NAS algorithms. This search space allows a mix of operations by composing different types of atomic blocks, while the search space in previous methods only allows homogeneous operations. Based on this search space, we propose a resource-aware architecture search framework which automatically assigns the computational resources (e.g., output channel numbers) for each operation by jointly considering the performance and the computational cost. In addition, to accelerate the search process, we propose a dynamic network shrinkage technique which prunes the atomic blocks with negligible influence on outputs on the fly. Instead of a search-and-retrain two-stage paradigm, our method simultaneously searches and trains the target architecture. Our method achieves state-of-the-art performance under several FLOPs configurations on ImageNet with a small searching cost. We open our entire codebase at: https://github.com/meijieru/AtomNAS.

## 1 INTRODUCTION

Human-designed neural networks are already surpassed by machine-designed ones. Neural Architecture Search (NAS) has become the mainstream approach to discover efficient and powerful network structures (Zoph & Le (2017); Pham et al. (2018); Tan et al. (2019); Liu et al. (2019a)). Although the tedious searching process is conducted by machines, humans still involve extensively in the design of the NAS algorithms. Designing of search spaces is critical for NAS algorithms and different choices have been explored. Cai et al. (2019) and Wu et al. (2019) utilize supernets with multiple choices in each layer to accommodate a sampled network on the GPU. Chen et al. (2019b) progressively grow the depth of the supernet and remove unnecessary blocks during the search. Tan & Le (2019a) propose to search the scaling factor of image resolution, channel multiplier and layer numbers in scenarios with different computation budgets. Stamoulis et al. (2019a) propose to use different kernel sizes in each layer of the supernet and reuse the weights of larger kernels for small kernels. Howard et al. (2019); Tan & Le (2019b) adopts Inverted Residuals with Linear Bottlenecks (MobileNetV2 block) (Sandler et al., 2018), a building block with light-weighted depth-wise convolutions for highly efficient networks in mobile scenarios.

However, the proposed search spaces generally have only a small set of choices for each block. DARTS and related methods (Liu et al., 2019a; Chen et al., 2019b; Liang et al., 2019) use around 10 different operations between two network nodes. Howard et al. (2019); Cai et al. (2019); Wu et al. (2019); Stamoulis et al. (2019a) search the expansion ratios in the MobileNetV2 block but still limit them to a few discrete values. We argue that search space of finer granularity is critical to find

---

* This work was done during the internship program at Bytedance.

optimal neural architectures. Specifically, the searched building block in a supernet should be as small as possible to generate the most diversified model structures.

We revisit the architectures of state-of-the-art networks (Howard et al. (2019); Tan & Le (2019b); He et al. (2016)) and discover a commonly used building structure: convolution - channel-wise operation - convolution. We reinterpret this building structure as an ensemble of computationally independent blocks, which we call *atomic blocks*. As the minimum search unit, the atomic block constitutes a much larger and more fine-grained search space, within which we are able to search for mixed operations (e.g., convolutions with different kernel sizes and their channel numbers).

For the efficient exploration of the new search space, we propose a NAS framework named Atom-NAS which applies network pruning techniques to architecture search. Specifically, we start from an initial large supernet and rewrite every convolution - channel-wise operation - convolution structure of it in the form the weighted sum of atomic blocks; the weights reflect the contribution of the atomic blocks to the network capacity and are called *importance factors*. For each atomic block, a penalty term in proportion to its FLOPs is enforced on its importance factor; effectively, the penalty makes AtomNAS favor atomic blocks with less FLOPs. By minimizing the combination of the original network loss and the total penalty on the weights, AtomNAS is able to learn both the parameters of the network and the weights of the atomic blocks. At the end of the learning, atomic blocks with very small weights (e.g., $< 0.001$) are removed from the network and we obtain the final network which has fewer FLOPs. Since the pruned atomic blocks have little contribution to the network output due to their negligible weights, the final network does not need to be retrained or finetuned.

Training on the large supernet is computationally demanding. We observe that for many pruned atomic blocks, their weights diminish at the early stage of learning and never "revive" throughout the rest of learning. We propose a dynamic network shrinkage technique which removes those atomic blocks on the fly and greatly reduces the run time of AtomNAS.

In our experiment, our method achieves 75.9% top-1 accuracy on ImageNet dataset around 360M FLOPs, which is 0.9% higher than state-of-the-art model (Stamoulis et al., 2019a). By further incorporating additional modules, our method achieves 77.6% top-1 accuracy. It outperforms MixNet by 0.6% using 363M FLOPs, which is a new state-of-the-art under the mobile scenario.

In summary, the major contributions of our work are:

1. We design a fine-grained search space which includes the exact number of channels and mixed operations (e.g., combination of different convolution kernels).

2. We propose an NAS framework, AtomNAS. Within the framework, an efficient end-to-end NAS algorithm is proposed which can simultaneously search the network architecture and train the final model. No finetuning is needed after the algorithm finishes.

3. With the proposed search space and AtomNAS, we achieve state-of-the-art performance on ImageNet dataset under mobile setting.

## 2 RELATED WORK

### 2.1 NEURAL ARCHITECTURE SEARCH

Recently, there is a growing interest in automated neural architecture design. Reinforce learning based NAS methods (Zoph & Le, 2017; Tan et al., 2019; Tan & Le, 2019b;a) are usually computational intensive, thus hampering its usage with limited computational budget. To accelerate the search procedure, ENAS (Pham et al., 2018) represents the search space using a directed acyclic graph and aims to search the optimal subgraph within the large supergraph. A training strategy of parameter sharing among subgraphs is proposed to significantly increase the searching efficiency. The similar idea of optimizing optimal subgraphs within a supergraph is also adopted by Liu et al. (2019a); Jin et al. (2019); Xu et al. (2020); Wu et al. (2019); Guo et al. (2019); Cai et al. (2019). Stamoulis et al. (2019a); Yu et al. (2020) further share the parameters of different paths within a block using super-kernel representation. A prominent disadvantage of the above methods is that their coarse search spaces only support selecting one out of a set of choices (e.g., selecting one kernel size from {3, 5, 7}). MixNet tries to benefit from mixed operations by using a predefined set of mixed operations {{3}, {3, 5}, {3, 5, 7}, {3, 5, 7, 9}}, where the channels are equally distributed

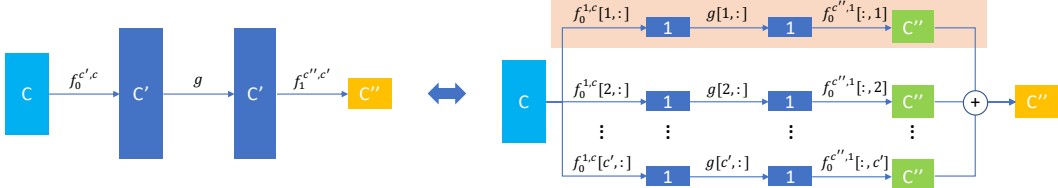

Figure 1: Illustration of the ensemble perspective. Arrow means operators. The structure of two convolutions joined by a channel-wise operation is mathematically equivalent to the ensemble of multiple atomic blocks, according to Eq. (2). Colored rectangles represent tensors, with numbers inside indicating their channel numbers; The shaded path on the right is one example of atomic block.

among different kernel sizes. Due to this limitation, it is difficult to learn optimal architectures under computational resource constraints. On the contrary, our method takes advantage of the fine-grained search space and is able to search for more flexible network architectures satisfying various resource constraints. The fine-grained search space proposed in this paper is exponentially larger than previous search space. For reference, the total number of possible structures within the experiment is around $10^{162}$, compared with $10^{21}$ for FBNet. Recently, to improve the final performance of the searched architectures, Yu et al. (2020) utilizes knowledge distillation which is orthogonal to our method. It could be easily integrated into our method by Eq. (5) thanks to the end-to-end learning paradigm of our method.

## 2.2 NETWORK PRUNING

Assuming that many parameters in the network are unnecessary, network pruning methods start from a computation-intensive model, identify the unimportant connections and remove them to get a compact and efficient network. Early method (Han et al., 2016) simultaneously learns the important connections and weights. However, non-regularly removing connections in these works makes it hard to achieve theoretical speedup ratio on realistic hardwares due to extra overhead in caching and indexing. To tackle this problem, structured network pruning methods (He et al., 2017b; Liu et al., 2017; Luo et al., 2017; Ye et al., 2018; Gordon et al., 2018) are proposed to prune structured components in networks, e.g. the entire channel and kernel. In this way, empirical acceleration can be achieved on modern computing devices. Liu et al. (2017); Ye et al. (2018); Gordon et al. (2018) encourage channel-level sparsity by imposing the L-1 regularizer on the channel dimension, which is also used by our method. Recently, Liu et al. (2019b) show that in structured network pruning, the learned weights are unimportant. This suggests structured network pruning is actually a neural architecture search focusing on channel numbers. Our method jointly searches the channel numbers and a mix of operations, which is a much larger search space.

## 3 ATOMNAS

We formulate our neural architecture search method in a fine-grained search space with the atomic block used as the basic search unit. An atomic block is comprised of two convolutions connected by a channel-wise operation. By stacking atomic blocks, we obtain larger building blocks (*e.g.* residual block and MobileNetV2 block proposed in a variety of state-of-the-art models including ResNet, MobileNet V2/V3 (He et al., 2016; Howard et al., 2019; Sandler et al., 2018). In Section 3.1, We first show larger network building blocks (*e.g.* MobileNetV2 block) can be represented by an ensembles of atomic blocks. Based on this view, we propose a fine-grained search space using atomic blocks. In Section 3.2, we propose a resource-aware atomic block selection method for end-to-end architecture search. Finally, we propose a dynamic network shrinkage technique in Section 3.3, which greatly reduces the search cost.

### 3.1 FINE-GRAINED SEARCH SPACE

Under the typical block-wise NAS paradigm (Tan et al., 2019; Tan & Le, 2019b), the search space of each block in a neural network is represented as the Cartesian product $\mathcal{C} = \prod_{i=1} \mathcal{P}_i$, where each $\mathcal{P}_i$ is the set of all choices of the $i$-th configuration such as kernel size, number of channels and type of operation. For example, $\mathcal{C} = \{\text{conv, depth-wise conv, dilated conv}\} \times \{3, 5\} \times \{24, 32, 64, 128\}$ represents a search space of three types of convolutions by two kernel sizes and four options of channel number. A block in the resulting model can only pick one convolution type from the three and one output channel number from the four values. This paradigm greatly limits the search space due to the few choices of each configuration. Here we present a more fine-grained search space by decomposing the network into smaller and more basic building blocks.

We denote $f^{c',c}(X)$ as a convolution operator, where $X$ is the input tensor and $c$, $c'$ are the input and output channel numbers respectively. A wide range of manually-designed and NAS architectures share a structure that joins two convolutions by a channel-wise operation:

$$Y = \left( f_1^{c'',c'} \circ g \circ f_0^{c',c} \right)(X) \tag{1}$$

where $g$ is a channel-wise operator. For example, in VGG (Simonyan & Zisserman, 2015) and a Residual Block (He et al., 2016), $f_0$ and $f_1$ are convolutions and $g$ is one of Maxpool, ReLU and BN-ReLU; in a MobileNetV2 block (Sandler et al., 2018), $f_0$ and $f_1$ are point-wise convolutions and $g$ is depth-wise convolution with BN-ReLU in the MobileNetV2 block. Eq. (1) can be reformulated as follows:

$$Y = \sum_{i=1}^{c'} \left( f_1^{c'',1}[i,:] \circ g[i,:] \circ f_0^{1,c}[:,i] \right)(X), \tag{2}$$

where $f_0^{1,c}[:,i]$ is the $i$-th convolution kernel of $f_0$, $g[i,:]$ is the operator of the $i$-th channel of $g$, and $\{f_1^{c'',1}[i,:]\}_{i=1}^{c'}$ are obtained by splitting the kernel tensor of $f_1$ along the the input channel dimension. Each term in the summation can be seen as a computationally independent block, which is called *atomic block*. Fig. (1) demonstrate this reformulation. By determining whether to keep each atomic block in the final model individually, the search of channel number $c'$ is enabled through channel selection, which greatly enlarges the search space.

This formulation also naturally includes the selection of operators. To gain a better understanding, we first generalize Eq. (2) as:

$$Y = \sum_{i=1}^{c'} \left( f_{1i}^{c'',1} \circ g_i \circ f_{0i}^{1,c} \right)(X). \tag{3}$$

Note the array indices $i$ are moved to subscripts. In this formulation, we can use different types of operators for $f_{0i}$, $f_{1i}$ and $g_i$; in other words, $f_0$, $f_1$ and $g$ can each be a combination of different operators and each atomic block can use different operators such as convolutions with different kernel sizes.

Formally, the search space is formulated as a supernet which is built based on the structure in Eq. (1); such structure satisfies Eq. (3) and thus can be represented by atomic blocks; each of $f_0$, $f_1$ and $g$ is a combination of operators. The new search space includes some state-of-the-art network architectures. For example, by allowing $g$ to be a combination of convolutions with different kernel sizes, the MixConv block in MixNet (Tan & Le, 2019b) becomes a special case in our search space. In addition, our search space facilitates discarding any number of channels in $g$, resulting in a more fine-grained channel configuration. In comparison, the channel numbers are determined heuristically in Tan & Le (2019b).

### 3.2 RESOURCE-AWARE ATOMIC BLOCK SEARCH

In this work, we adopt a differentiable neural architecture search paradigm where the model structure is discovered in a full pass of model training. With the supernet defined above, the final model can be produced by discarding part of the atomic blocks during training. Following DARTS (Liu et al.

(2019a)), we introduce a importance factor $\alpha$ to scale the output of each atomic block in the supernet. Eq. (3) then becomes

$$Y = \sum_{i=1}^{c'} \alpha_i \left( f_{1i}^{c'',1} \circ g_i \circ f_{0i}^{1,c} \right) (X). \tag{4}$$

Here, each $\alpha_i$ is tied with an atomic block comprised of three operators $f_{1i}^{c'',1}$, $g_i$ and $f_{0i}^{1,c}$. The importance factors are learned jointly with the network weights. Once the training finishes, the atomic blocks that have negligible effect (i.e., those with factors smaller than a threshold) on the network output are discarded.

We still need to address two issues related to the importance factors $\alpha_i$'s. The first issue is where in the supernet we should put the $\alpha$? Let's first consider the case when $g$ only contains linear operations, e.g., convolution, batch normalization and linear activation like ReLU. If $g$ contains at least one BN layer, The scaling parameters in the BN layers can be directly used as such importance factors (Liu et al. (2017)). If $g$ has no BN layers, which is rare, we can place $\alpha$ anywhere between $f_0$ and $f_1$; however, we need to apply regularization terms to the weights of $f_0$ and $f_1$ (e.g., weight decays) in order to prevent weights in $f_0$ and $f_1$ from getting too large and canceling the effect of $\alpha$. When $g$ contains non-linear operations, e.g., Swish activation and Sigmoid activation, we can only put $\alpha$ behind $f_1$.

The second issue is how to avoid performance deterioration after discarding some of the atomic blocks. For example, DARTS discards operations with small scale factors after iterative training of model parameters and scale factors. Since the scale factors of the discarded operations are not small enough, the performance of the network will be affected which needs re-training to adjust the weights again. In order to maintain the performance of the supernet after dropping some atomics blocks, the importance factors $\alpha$ of those atomic blocks should be sufficiently small. Inspired by the channel pruning work in Liu et al. (2017), we add L1 norm penalty loss on $\alpha$, which effectively pushes many importance factors to near-zero values. At the end of learning, atomic blocks with $\alpha$ close to zero are removed from the supernet. Note that since the BN scales change more dramatically during training due to the regularization term, the running statistics of BNs might be inaccurate and needs to be calculated again using the training set.

With the added regularization term, the training loss is

$$\mathcal{L} = \mathcal{E} + \lambda \sum_{i \in \mathcal{S}} c_i |\alpha_i|, \tag{5}$$

$$c_i = \hat{c}_i / \sum_{k \in \mathcal{S}} \hat{c}_k \tag{6}$$

where $\lambda$ is the coefficient of L1 penalty term, $\mathcal{S}$ is the index set of all atomic blocks, and $\mathcal{E}$ is the conventional training loss (e.g., cross-entropy loss combined with the regularization term like weight decay and distillation loss.). $|\alpha_i|$ is weighted by coefficient $c_i$ which is proportional to the computation cost of $i$-th atomic block, i.e. $\hat{c}_i$. By using computation costs aware regularization, we encourage the model to learn network structures that strike a good balance between accuracy and efficiency. In this paper, we use FLOPs as the criteria of computation cost. Other metrics such as latency and energy consumption can be used similarly. As a result, the whole loss function $\mathcal{L}$ trades off between accuracy and FLOPs.

### 3.3 Dynamic Network Shrinkage

Usually, the supernet is much larger than the final search result. We observe that many atomic blocks become "dead" starting from the early stage of the search, i.e., their importance factors $\alpha$ are close to zero till the end of the search. To utilize computational resources more efficiently and speed up the search process, we propose a dynamic network shrinkage algorithm which cuts down the network architecture by removing atomic blocks once they are deemed "dead".

We adopt a conservative strategy to decide whether an atomic block is "dead": for importance factors $\alpha$, we maintain its momentum $\hat{\alpha}$ which is updated as

$$\hat{\alpha} \leftarrow \beta \hat{\alpha} + (1 - \beta) \alpha^t, \tag{7}$$

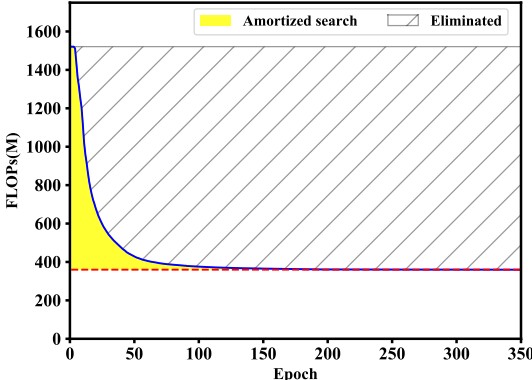

Figure 2: FLOPs change of the supernet during the searching and training for AtomNAS-C. The crossed-out region corresponds to the saved computation compared to training the supernet without the dynamic shrinkage. The region in yellow corresponds to the extra cost compared with training the final model from scratch, the cost of which is the region below the red dashed line.

---

Initialize the supernet and the exponential moving average;
**while** $epoch \leq max\_epoch$ **do**
  Update network weights and importance factors $\alpha$ by minimizing the loss function $\mathcal{L}$ ;
  Update the $\hat{\alpha}$ by Eq. (7);
  **if** *Total FLOPs of dead blocks* $\geq \Delta$ **then**
    Remove dead blocks from the supernet;
  **end**
  Recalculate BN's statistics by forwarding some training examples;
  Validate the performance of the current supernet;
**end**

**Algorithm 1:** Dynamic network shrinkage

---

where $\alpha^t$ is the importance factors at $t$-th iteration and $\beta$ is the decay term. An atomic block is considered "dead" if both $\hat{\alpha}$ and $\alpha^t$ are smaller than a threshold, which is set to $0.001$ throughout experiments.

Once the total FLOPs of "dead" blocks reach a predefined threshold, we remove those blocks from the supernet. As discussed above, we recalculate BN's running statistics before deploying the network. The whole training process is presented in Algorithm 1.

We show the FLOPs of a sample network during the search process in Fig. 2. We start from a supernet with 1521M FLOPs and dynamically discard "dead" atomic blocks to reduce search cost. The overall search and train cost only increases by $17.2\%$ compared to that of training the searched model from scratch.

## 4 EXPERIMENT

We first describe the implementation details in Section 4.1 and then compare AtomNAS with previous state-of-the-art methods under various FLOPs constraints in Section 4.2. In Section 4.3, we provide more detailed analysis about AtomNAS. Finally, in Section 4.4, we demonstrate the transferability of AtomNAS networks by evaluating them on detection and instance segmentation tasks.

### 4.1 IMPLEMENTATION DETAILS

The architecture of the supernet we use for the experiments is shown in table on the right of Fig. 3. The supernet contains 21 AtomNAS blocks, the searchable block in our supernet; the picture on the right of Fig. 3 illustrates the structure of an AtomNAS block, where $f_0$ is a $1 \times 1$ pointwise convolutions that expands the input channel number from $C$ to $3 \times 6C$; $g$ is a mix of three depth-wise

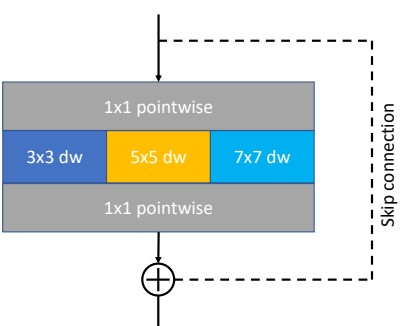

| Input Shape | Block | f | n | stride |
|---|---|---|---|---|
| $224^2 \times 3$ | 3x3 conv | 32(16) | 1 | 2 |
| $112^2 \times 32(16)$ | 3x3 MB | 16 | 1 | 1 |
| $112^2 \times 16$ | searchable | 24 | 4 | 2 |
| $56^2 \times 24$ | searchable | 40 | 4 | 2 |
| $28^2 \times 40$ | searchable | 80 | 4 | 2 |
| $14^2 \times 80$ | searchable | 96 | 4 | 1 |
| $14^2 \times 96$ | searchable | 192 | 4 | 2 |
| $7^2 \times 192$ | searchable | 320 | 1 | 1 |
| $7^2 \times 320$ | avgpool | - | 1 | 1 |
| 1280 | fc | 1000 | 1 | - |

Figure 3: (Left) The searchable block of the supernet. $f_0$ and $f_1$ are fixed to $1 \times 1$ pointwise convolutions; $g$ here is a mix of three convolutions with kernel sizes of $3 \times 3$, $5 \times 5$ and $7 \times 7$. $f_0$ expands the input channel number from $C$ to $18C$ and $f_1$ projects the channel number to the output channel number. If the output dimension stays the same as the input dimension, we use a skip connection to add the input to the output. (Right) Architecture of the supernet. Column-Block denotes the block type; MB denotes MobileNetV2 block; "searchable" means a searchable block shown on the left. Column-f denotes the output channel number of a block. Column-n denotes the number of blocks. Column-s denotes the stride of the first block in a stage. The output channel numbers of the first convolution are 16 for AtomNAS-A, 32 for AtomNAS-B and AtomNAS-C.

convolutions with kernel sizes of $3 \times 3$, $5 \times 5$ and $7 \times 7$, and $f_1$ is another $1 \times 1$ pointwise convolutions that projects the channel number to the output channel number. Similar to MobileNetV2 (Sandler et al., 2018), if the output dimension stays the same as the input dimension, we use a skip connection to add the input to the output. AtomNAS block is effectively an ensemble of $3 \times 6C$ atomic blocks, whose underlying search space covers the MobileNetV2 block (Sandler et al., 2018) and its multi-kernel variant, MixConv (Tan & Le, 2019b). Within AtomNAS block, we are able to optimize the distribution of computation resources (i.e., channel numbers) among the three types of depth-wise convolution.

We use the same training configuration (e.g., RMSProp optimizer, EMA on weights and exponential learning rate decay) as Tan et al. (2019); Stamoulis et al. (2019a) and do not use extra data augmentation such as MixUp (Zhang et al., 2018) and AutoAugment (Cubuk et al., 2018). We find that using this configuration is sufficient for our method to achieve good performance. Our results are shown in Table 1 and Table 3. When training the supernet, we use a total batch size of 2048 on 32 Tesla V100 GPUs and train for 350 epochs. For our dynamic network shrinkage algorithm, we set the momentum factor $\beta$ in Eq. (7) to 0.9999. At the beginning of the training, all of the weights are randomly initialized. To avoid removing atomic blocks with high penalties (i.e., FLOPs) prematurely, the weight of the penalty term in Eq. (5) is increased from 0 to the target $\lambda$ by a linear scheduler during the first 25 epochs. By setting the weight of the L1 penalty term $\lambda$ to be $1.8 \times 10^{-4}$, $1.2 \times 10^{-4}$ and $1.0 \times 10^{-4}$ respectively, we obtain networks with three different sizes: AtomNAS-A, AtomNAS-B, and AtomNAS-C. They have the similar FLOPs as previous state-of-the-art networks under 400M: MixNet-S (Tan & Le, 2019b), MixNet-M (Tan & Le, 2019b) and SinglePath (Stamoulis et al., 2019a). In Appendix A, we visualize the architecture of AtomNAS-C.

## 4.2 EXPERIMENTS ON IMAGENET

We apply AtomNAS to search high performance light-weight model on ImageNet 2012 classification task (Deng et al., 2009). Table 1 compares our methods with previous state-of-the-art models, either manually designed or searched.

With models directly produced by AtomNAS, our method achieves the new state-of-the-art under all FLOPs constraints. Especially, AtomNAS-C achieves 75.9% top-1 accuracy with only 360M FLOPs, and surpasses all other models, including models like PDARTS and DenseNAS which have much higher FLOPs.

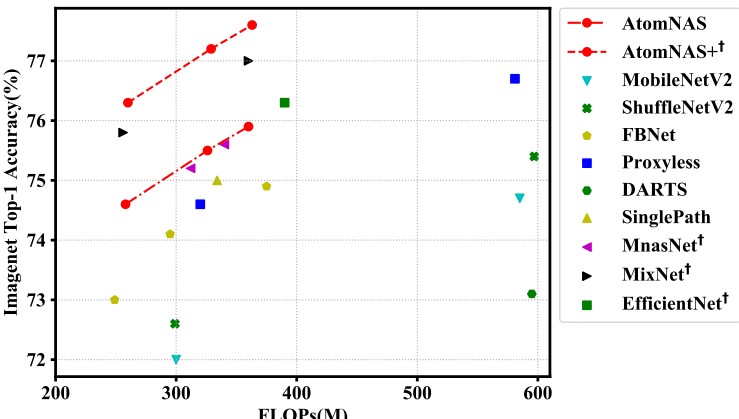

Figure 4: FLOPs versus accuracy on ImageNet. $^{\dagger}$ means methods use extra techniques like Swish activation and Squeeze-and-Excitation module.

Techniques like Swish activation function (Ramachandran et al., 2018) and Squeeze-and-Excitation (SE) module (Hu et al., 2018) consistently improve the accuracy with marginal FLOPs cost. For a fair comparison with methods that use these techniques, we directly modify the searched network by replacing all ReLU activation with Swish and add SE module with ratio 0.5 to every block and then retrain the network from scratch. Note that unlike other methods, we do not search the configuration of Swish and SE, and therefore the performance might not be optimal. Extra data augmentations such as MixUp and AutoAugment are still not used. We train the models from scratch with a total batch size of 4096 on 32 Tesla V100 GPUs for 250 epochs.

Simply adding these techniques improves the results further. AtomNAS-A+ achieves 76.3% top-1 accuracy with 260M FLOPs, which outperforms many heavier models including MnasNet-A2. Without extra data augmentations, it performs as well as Efficient-B0 (Tan & Le, 2019a) by using 130M less FLOPs. It also outperforms the previous state-of-the-art MixNet-S by 0.5%. In addition, AtomNAS-C+ improves the top-1 accuracy on ImageNet to 77.6%, surpassing previous state-of-the-art MixNet-M by 0.6% and becomes the overall best performing model under 400M FLOPs.

Fig. 4 visualizes the top-1 accuracy on ImageNet for different models. It's clear that our fine-grained search space and the end-to-end resource-aware search method boost the performance significantly.

### 4.3 ANALYSIS

#### 4.3.1 RESOURCE-AWARE REGULARIZATION

To demonstrate the effectiveness of the resource-aware regularization in Section 3.2, we compare it with a baseline without FLOPs-related coefficients $c_i$, which is widely used in network pruning (Liu et al., 2017; He et al., 2017b). Table 2 shows the results. First, by using the same L1 penalty coefficient $\lambda = 1.0 \times 10^{-4}$, the baseline achieves a network with similar performance but using much more FLOPs; then by increasing $\lambda$ to $1.5 \times 10^{-4}$, the baseline obtain a network which has similar FLOPs but inferior performance (i.e., about 1.0% lower). In Fig. 6b we visualized the ratio of different types of atomic blocks of the baseline network obtained by $\lambda = 1.5 \times 10^{-4}$. The baseline network keeps more atomic blocks in the earlier blocks, which have higher computation cost due to higher input resolution. On the contrary, AtomNAS is aware of the resource constraint, thus keeping more atomic blocks in the later blocks and achieving much better performance.

#### 4.3.2 BN RECALIBRATION

As the BN's running statistics might be inaccurate as explained in Section 3.2 and Section 3.3, we re-calculate the running statistics of BN before inference, by forwarding 131k randomly sampled training images through the network. Table 3 shows the impact of the BN recalibration. The top-1 accuracies of AtomNAS-A, AtomNAS-B, and AtomNAS-C on ImageNet improve by 1.4%, 1.7%, and 1.2% respectively, which clearly shows the benefit of BN recalibration.

Table 1: Comparision with state-of-the-arts on ImageNet under the mobile setting. † denotes methods using extra network modules such as Swish activation and Squeeze-and-Excitation module. ‡ denotes using extra data augmentation such as MixUp and AutoAugment. * denotes models searched and trained simultaneously.

| Model | Parameters | FLOPs | Top-1(%) | Top-5(%) |
|---|---|---|---|---|
| MobileNetV1 (Howard et al., 2017) | 4.2M | 575M | 70.6 | 89.5 |
| MobileNetV2 (Sandler et al., 2018) | 3.4M | 300M | 72.0 | 91.0 |
| MobileNetV2 (our impl.) | 3.4M | 301M | 73.6 | 91.5 |
| MobileNetV2 (1.4) | 6.9M | 585M | 74.7 | 92.5 |
| ShuffleNetV2 (Ma et al., 2018) | 3.5M | 299M | 72.6 | - |
| ShuffleNetV2 2× | 7.4M | 591M | 74.9 | - |
| FBNet-A (Wu et al., 2019) | 4.3M | 249M | 73.0 | - |
| FBNet-C | 5.5M | 375M | 74.9 | - |
| Proxyless (mobile) (Cai et al., 2019) | 4.1M | 320M | 74.6 | 92.2 |
| SinglePath (Stamoulis et al., 2019a) | 4.4M | 334M | 75.0 | 92.2 |
| NASNet-A (Zoph & Le, 2017) | 5.3M | 564M | 74.0 | 91.6 |
| DARTS (second order) (Liu et al., 2019a) | 4.9M | 595M | 73.1 | - |
| PDARTS (cifar 10) (Chen et al., 2019b) | 4.9M | 557M | 75.6 | 92.6 |
| DenseNAS-A (Fang et al., 2019) | 7.9M | 501M | 75.9 | 92.6 |
| FairNAS-A (Chu et al., 2019b) | 4.6M | 388M | 75.3 | 92.4 |
| **AtomNAS-A*** | 3.9M | 258M | 74.6 | 92.1 |
| **AtomNAS-B*** | 4.4M | 326M | 75.5 | 92.6 |
| **AtomNAS-C*** | 4.7M | 360M | 75.9 | 92.7 |
| SCARLET-A† (Chu et al., 2019a) | 6.7M | 365M | 76.9 | 93.4 |
| MnasNet-A1† (Tan et al., 2019) | 3.9M | 312M | 75.2 | 92.5 |
| MnasNet-A2† | 4.8M | 340M | 75.6 | 92.7 |
| MixNet-S† (Tan & Le, 2019b) | 4.1M | 256M | 75.8 | 92.8 |
| MixNet-M† | 5.0M | 360M | 77.0 | 93.3 |
| EfficientNet-B0†‡ (Tan & Le, 2019a) | 5.3M | 390M | 76.3 | 93.2 |
| SE-DARTS+†‡ (Liang et al., 2019) | 6.1M | 594M | 77.5 | 93.6 |
| **AtomNAS-A+**† | 4.7M | 260M | 76.3 | 93.0 |
| **AtomNAS-B+**† | 5.5M | 329M | 77.2 | 93.5 |
| **AtomNAS-C+**† | 5.9M | 363M | 77.6 | 93.6 |

Table 2: Influence of awareness of resource metric. The upper block uses equal penalties for all atomic blocks. The lower part uses our resource-aware atomic block selection.

| $\lambda$ | FLOPs | Top-1(%) |
|---|---|---|
| $1.0 \times 10^{-4}$ | 445M | 76.1 |
| $1.5 \times 10^{-4}$ | 370M | 74.9 |
| $1.0 \times 10^{-4}$ | 360M | 75.9 |

### 4.3.3 COST OF DYNAMIC NETWORK SHRINKAGE

Our dynamic network shrinkage algorithm speedups the search and train process significantly. For AtomNAS-C, the total time for search-and-training is 25.5 hours. For reference, training the final architecture from scratch takes 22 hours. Note that as the supernet shrinks, both the GPU memory consumption and forward-backward time are significantly reduced. Thus it's possible to dynamically change the batch size once having sufficient GPU memory, which would further speed up the whole procedure.

Table 3: Influence of BN recalibration.

| Model | w/o Recalibration | w/ Recalibration |
|---|---|---|
| AtomNAS-A | 73.2 | 74.6 (+1.4) |
| AtomNAS-B | 73.8 | 75.5 (+1.7) |
| AtomNAS-C | 74.7 | 75.9 (+1.2) |

## 4.4 EXPERIMENTS ON COCO DETECTION AND INSTANCE SEGMENTATION

In this section, we assess the performance of AtomNAS models as feature extractors for object detection and instance segmentation on COCO dataset (Lin et al., 2014). We first pretrain AtomNAS models (without Swish activation function (Ramachandran et al., 2018) and Squeeze-and-Excitation (SE) module (Hu et al., 2018)) on ImageNet, use them as drop-in replacements for the backbone in the Mask-RCNN model (He et al., 2017a) by building the detection head on top of the last feature map, and finetune the model on COCO dataset.

We use the open-source code MMDetection (Chen et al., 2019a). All the models are trained on COCO train2017 with batch size 16 and evaluated on COCO val2017. Following the schedule used in the open-source implementation of TPU-trained Mask-RCNN , the learning rate starts at 0.02 and decreases by a scale of 10 at 15-th and 20th epoch respectively. The models are trained for 23 epochs in total.

Table 4 compares the results with other baseline backbone models. The detection results of baseline models are from Stamoulis et al. (2019b). We can see that all three AtomNAS models outperform the baselines on object detection task. The results demonstrate that our models have better transferability than the baselines, which may due to mixed operations, a.k.a multi-scale here, are more important to object detection and instance segmentation.

Table 4: Comparision with baseline backbones on COCO object detection and instance segmentation. Cls denotes the ImageNet top-1 accuracy; detect-mAP and seg-mAP denotes mean average precision for detection and instance segmentation on COCO dataset. The results of baseline models are from Stamoulis et al. (2019b). SinglePath+ (Stamoulis et al., 2019b) contains SE module.

| Model | FLOPs | Cls (%) | detect-mAP (%) | seg-mAP (%) |
|---|---|---|---|---|
| MobileNetV2 (Sandler et al., 2018) | 301M | 73.6 | 30.5 | - |
| Proxyless (mobile) (Cai et al., 2019) | 320M | 74.6 | 32.9 | - |
| Proxyless (mobile) (our impl.) | 320M | 74.9 | 32.7 | 30.0 |
| SinglePath+ (Stamoulis et al., 2019b) | 353M | 75.6 | 33.0 | - |
| SinglePath (our impl.) | 334M | 75.0 | 32.0 | 29.7 |
| **AtomNAS-A** | 258M | 74.6 | 32.7 | 30.1 |
| **AtomNAS-B** | 326M | 75.5 | 33.6 | 30.8 |
| **AtomNAS-C** | 360M | 75.9 | 34.1 | 31.4 |

## 5 CONCLUSION

In this paper, we revisit the common structure, i.e., two convolutions joined by a channel-wise operation, and reformulate it as an ensemble of atomic blocks. This perspective enables a much larger and more fine-grained search space. For efficiently exploring the huge fine-grained search space, we propose an end-to-end framework named AtomNAS, which conducts architecture search and network training jointly. The searched networks achieve significantly better accuracy than previous state-of-the-art methods while using small extra cost.

---

https://github.com/tensorflow/tpu/tree/master/models/official/mask_rcnn

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

## A    VISUALIZATION

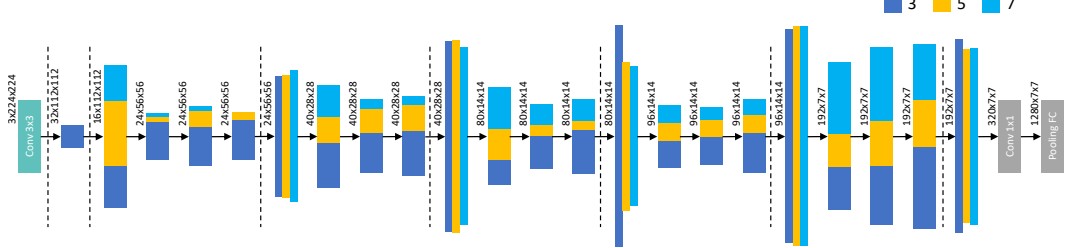

Figure 5: The architecture of AtomNAS-C. Blue, orange, cyan blocks denote atomic blocks with kernel size 3, 5 and 7 respectively; the heights of these blocks are proportional to their expand ratios.

We plot the structure of the searched architecture AtomNAS-C in Fig. 5, from which we see more flexibility of channel number selection, not only among different operators within each block, but also across the network. In Fig. 6a, we visualize the ratio between atomic blocks with different kernel sizes in all 21 search blocks. First, we notice that all search blocks have convolutions of all three kernel sizes, showing that AtomNAS learns the importance of using multiple kernel sizes in network architecture. Another observation is that AtomNAS tends to keep more atomic blocks at the later stage of the network. This is because in earlier stage, convolutions of the same kernel size costs more FLOPs; AtomNAS is aware of this (thanks to its resource-aware regularization) and try to keep as less as possible computationally costly atomic blocks.

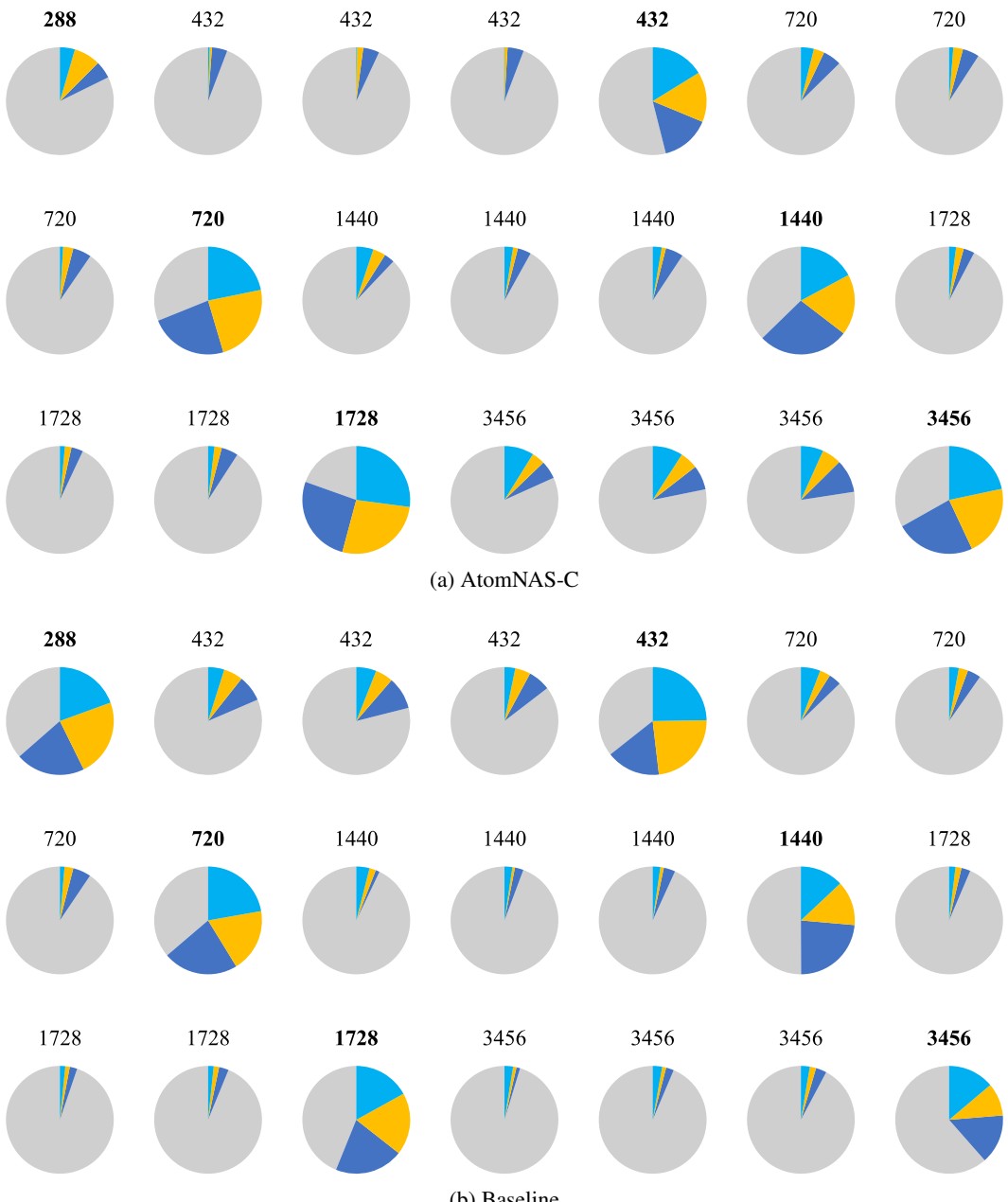

Figure 6: Ratio of different types of atomic blocks in all 21 searchable blocks. The text above each pie tells the total number of atomic blocks of the corresponding block in the original supernet. Grey denotes dead atomic blocks; blue, orange, and cyan represent atomic blocks using depth-wise convolutions with kernel size $3, 5, 7$ respectively. Blocks without skip connection are highlighted by bold text. (a) Visualization for AtomNAS-C. (b) Visualization for baseline (i.e., without FLOPs related coefficients $c_i$).

