# OpenReview forum: "AtomNAS: Fine-Grained End-to-End Neural Architecture Search"
_ICLR.cc/2020/Conference — Accept (Poster)_

### Official Review · AnonReviewer3 · 2019-10-20
**Official Blind Review #3**

**Rating:** 6

**Review:**

The authors propose AtomNAS, a neural architecture search (NAS) algorithm, with a new fine-grained search space and the dynamic network shrinkage method. The searched models achieve a new state-of-the-art result on the ImageNet classification task for mobile setting with restricted FLOPs.

- The proposed method is novel and technically sound. In addition, the experimental results on ImageNet are impressive. However, the experiment section is not solid enough.

- The authors do not include the searching cost and inference latency in Table 1. Different NAS papers have different objectives and searching cost. For instance, ProxylessNAS (Cai et al., 2019) and DenseNAS (Fang et al., 2019) focus on searching cost. They require only 200 and 92 GPU hours (with TITAN XP). However, the proposed AtomNAS takes 32 * 25.5 = 816 GPU hours (with V100). The authors only point out that DenseNAS uses more parameters.  It would be better if the authors can make the comparison more transparent.

- I wonder when given the same searching budgets as ProxylessNAS and DenseNAS, how well AtomNAS can perform.

- The authors use only one dataset: ImageNet. I would like to see results on some other datasets or tasks. For instance, the authors may apply AtomNAS to other image classification datasets or finetuning the pre-trained models on object detection or semantic segmentation tasks.

- In general, the paper is well written and easy to follow. I would encourage the authors to add legends to Figure 5 and Figure 6. While the meaning of each color is explained in the caption, it is not straight forward.

In short, the proposed method is interesting and the results on ImageNet are impressive, I weakly accept this paper and hope that the authors can make the experiment section more solid in a revised version.


**Experience Assessment:**

I have read many papers in this area.

**Review Assessment: Checking Correctness Of Derivations And Theory:**

N/A

**Review Assessment: Checking Correctness Of Experiments:**

I carefully checked the experiments.

**Review Assessment: Thoroughness In Paper Reading:**

I read the paper at least twice and used my best judgement in assessing the paper.

---

> ### Author Response · Authors · 2019-11-15
> **Authors' Reply to Review #3**
>
> We thank the reviewer for appreciating our method. We respond to each concern below.
>
> Q: "The authors do not include the searching cost and inference latency in Table 1...The authors only point out that DenseNAS uses more parameters.  It would be better if the authors can make the comparison more transparent."
>
> "I wonder when given the same searching budgets as ProxylessNAS and DenseNAS, how well AtomNAS can perform."
>
> First, we would like to clarify that the search cost of ProxylessNAS and DenseNAS are measured on V100, not Titan XP, as confirmed by the authors. Titan XP was used to measure the latency in DenseNAS paper.
>
> Our method simultaneously search and train the final architecture in an end-to-end manner; the result model can be directly used for inference.  On the other hand, previous NAS methods like ProxylessNAS and DenseNAS require retraining the searched architecture from scratch after searching. There is no direct way to measure the search cost of our method, as the search and the training are mixed. For reference, simply training the searched architecture from scratch takes 704 V100 hours, and thus the “extra” cost of our method is around 112(=816-704) V100 hours. The cost is much less than ProxylessNAS and 20 V100 hours more than DenseNAS. However, DenseNAS uses a small portion of the original ImageNet training data (quote from the paper “For the search process, we randomly choose 100 classes from the original 1000 classes of the ImageNet training set. We sample 20% data in each class of the ImageNet subset to form the validation dataset. The remaining data is used for training.”); in other words, DenseNAS uses a proxy dataset, unlike ours and ProxylessNAS which both use the original data.
>
> At last, we would like to thank Review 3 for bringing up this discussion, as we are definitely interested in further reducing the computational complexity of our method.
>
> Q: The authors use only one dataset: ImageNet. I would like to see results on some other datasets or tasks. For instance, the authors may apply AtomNAS to other image classification datasets or finetuning the pre-trained models on object detection or semantic segmentation tasks.
>
> See 'Reply To Common Concerns' above.

---

### Official Review · AnonReviewer2 · 2019-10-22
**Official Blind Review #2**

**Rating:** 6

**Review:**

[Summary] This paper proposes a channel-wise neural architecture search (NAS) approach. The NAS search algorithm is similar to previous one-shot NAS, but the search space is channel-wise: each channel has it’s own kernel size, which is quite novel and interesting. Results are strong in terms of FLOPS and parameters.

[High-level comments]:

1. I like the novel idea of channel-wise search space, which provides great flexibility for NAS.  Although some recent works (e.g., MixNet) have tried to partition channels into groups, this paper goes further and searches for different kernel size for each single channel.  With this channel-wise search space, it naturally enables a combination of per-channel kernel size selection and per-channel pruning, leading to strong results in terms of FLOPS and parameters, as shown in Figure 4 and Table 1.

2. In general, channel-wise NAS is difficult as different channels are often coupled in various ways. However, the authors observe that recent NAS (such as MnasNet/MixNet/SCARLET-A/EfficientNet) are mostly based on a common MB pattern. By targeting to this specific pattern and applying some additional constraints (e.g. fixed input/output channel size and fixed number of layers per stage as shown in Figure 3), the authors successfully make the channel-wise NAS work well.  I appreciate the authors efforts, but I am also a little concerned that the proposed approach might be limited to this specific small search space.

[Questions and suggestions to authors]:

3.  How do you justify the generality of the channel-wise search space?  For example, is it possible to also search for input/output channel size (column f in Figure 3) and #layers per stage (column n in Figure 3)? Adding some discussions for this would be very helpful.

4. The title seems too broad. I recommend the authors including “channel-wise” in the title.

5. Please provide some justifications on how to set λ  and c_i in Equation (5).

6. When you say “expands the input channel number from C to 3 × 6C”, what does “3x6C” mean? Is it 18C? What’s the reason for choosing this specific value?

7. Could you show the accuracy and complexity (either FLOPS or params) of the supernet during the training? This information would be helpful to interpret and justify your algorithm 1.

8. The network architecture in Figure 5 is vague. Could you provide the network source code or frozen graph for this specific model?


**Experience Assessment:**

I have published in this field for several years.

**Review Assessment: Checking Correctness Of Derivations And Theory:**

I assessed the sensibility of the derivations and theory.

**Review Assessment: Checking Correctness Of Experiments:**

I assessed the sensibility of the experiments.

**Review Assessment: Thoroughness In Paper Reading:**

I read the paper at least twice and used my best judgement in assessing the paper.

---

> ### Author Response · Authors · 2019-11-15
> **Authors' Reply to Review #2**
>
> We thank the reviewer for the detailed comments. Below we provide responses to each concern.
>
> 3. How do you justify the generality of the channel-wise search space?  For example, is it possible to also search for input/output channel size (column f in Figure 3) and #layers per stage (column n in Figure 3)? Adding some discussions for this would be very helpful.
>
> Our formulation (Equation (1)-(3)) is not only for MobileNetV2 block, but also compatible with other structures like conv-conv, conv-maxpool-conv and  conv-bn-relu-conv, which are the building blocks of many popular architectures (e.g., VGG Net, Residual block). We use the MobileNetV2 blocks in the experiments as its wide application in state-of-the-art NAS methods thus easing the comparison between our method and them.
>
> We don’t introduce additional constraints. Most NAS methods (e.g., [1,2,3,4]) specify the skeleton of the supernet where the input/output channel is manually determined. It’s an interesting research topic in the NAS community. As for “#layers per stage”, it is actually allowed to change during the search process: when all atomic blocks within a layer are removed, the layer is essentially a skip connection. Although in practice, this never happens.
>
> At last, we want to emphasize that our method has an exponentially larger and more flexible search space than previous methods. The biggest contribution of our method is that we could use mixed operations instead of selecting one operation from a few options as did in previous methods. In this way, the search space is much more flexible and bigger than the previous ones. For your reference, the total number of possible structures within the experiment is around $10e162$, compared with $10e21$ for FBnet. It’s straightforward to extend our method into using mixed convolution types, mixed activation functions, and so on.
>
> 4. The title seems too broad. I recommend the authors including “channel-wise” in the title.
>
> Thanks for the advice. We will figure out a more proper title.
>
> 5. Please provide some justifications on how to set $\lambda$  and $c_i$ in Equation (5).
>
> As the λ increases, the FLOPs of the final model decrease. We set $\lambda$ in a heuristic way, such that the resulting models have similar FLOPs as previous state-of-the-art networks under 400M: MixNet-S [1], MixNet-M [1] and SinglePath [4].
>
> c_i’s are computed by equation 6, where we first calculate the FLOPs of every atomic block in the model and then normalize them globally to get $c_i$.
>
> 6. When you say “expands the input channel number from C to $3\times 6$C”, what does “$3\times 6$C” mean? Is it 18C? What’s the reason for choosing this specific value?
>
> It means 18C. We choose this value as 6 is widely used as the maximum expansion ratio [1, 2, 3].
>
> 7. Could you show the accuracy and complexity (either FLOPS or params) of the supernet during the training? This information would be helpful to interpret and justify your algorithm 1.
>
> The top-1 accuracy of the supernet is 78.39. As mentioned in Section 3.3, the FLOPS of supernet is 1521M with 11M parameters.
>
> 8. The network architecture in Figure 5 is vague. Could you provide the network source code or frozen graph for this specific model?
>
> We have released the code (including search). See 'Reply To Common Concerns' above.
>
> ===References===
> [1] M. Tan, et al., Mnasnet: Platform-aware neural architecture search for mobile. In CVPR, 2019.
> [2] H. Cai, et al., Proxylessnas: Direct neural architecture search on target task
> and hardware. In ICLR, 2019.
> [3] B. Wu, et al., Fbnet: Hardware-aware efficient convnet design via
> differentiable neural architecture search. In CVPR, 2019.
> [4] Stamoulis, et al., Single-path NAS: designing hardware-efficient convnets in less than 4hours. In CoRR, 2019.

---

### Official Review · AnonReviewer1 · 2019-10-23
**Official Blind Review #1**

**Rating:** 3

**Review:**

This basic idea of this paper is to decompose the common building blocks of large network into atomic blocks, which equips NAS with more fine-grained search space. What's more, the authors propose a resource-aware search to reduce the computation and dynamically shrinkage the model to accelerate the learning. Retraining the final network is no longer needed. They achieve state of art on ImageNet under several complexity constraints.

Pros:
Novel idea: the insight of this paper is that "larger network building blocks can be represented by an ensemble of atomic blocks". With this in hand, it can search the exact channel number through channel selection (i.e. atomic block selection, according to my understanding).

Efficiency: Resource-aware selection and dynamical shrinkage of the model also make it more efficient in inference and training.

Cons:
It would be better if the author could provide some comparison on GPU time. Since FLOPs is only an indirect metric for speed evaluation.

The biggest problem of this paper is that experiment is not enough. It would be more convincing if experiments on other popular datasets (CIFAR10/100 etc.) or tasks (object detection, semantic segmentation, etc.) are implemented.

Conclusion:
This is an interesting paper with novel idea and efficient implementation. However, more experiments are needed to validate the utility of the proposed method.

**Experience Assessment:**

I do not know much about this area.

**Review Assessment: Checking Correctness Of Derivations And Theory:**

N/A

**Review Assessment: Checking Correctness Of Experiments:**

I carefully checked the experiments.

**Review Assessment: Thoroughness In Paper Reading:**

I read the paper at least twice and used my best judgement in assessing the paper.

---

> ### Author Response · Authors · 2019-11-15
> **Authors' Reply to Review #1**
>
> We thank the reviewer for the detailed comments. Below we provide responses to each concern.
>
> Q: It would be better if the author could provide some comparison on GPU time. Since FLOPs is only an indirect metric for speed evaluation.
>
> The running time highly depends on implementation details (e.g., GPU/CPU, CUDA version, deep learning framework, etc.). For example, each block in our architecture has a mix of three different conv kernels; we haven’t apply any optimization to it (e.g., paralleling the computation of three kernels). Under the same total number of kernels, our block is slower than those with a single kernel size, e.g., a typical MobileNet V2 block. In addition, our method specifically targets at reducing the FLOPs, not latency. Therefore FLOPs is a fairer metric than inference speed like GPU time at this stage.
>
> Q: The biggest problem of this paper is that experiment is not enough. It would be more convincing if experiments on other popular datasets (CIFAR10/100 etc.) or tasks (object detection, semantic segmentation, etc.) are implemented.
>
> See 'Reply To Common Concerns' above.

---

### Author Response · Authors · 2019-11-15
**Reply To Common Concerns**

We appreciate the invaluable comments from the reviewers. Below is our response to the common concerns and questions from all reviewers.

- Code Release

We have released the whole codebase including search, which could be accessed with the following links:
https://anonymous.4open.science/r/ced78872-1992-43b9-ad69-2d611a14616d/

- Transfer learning on COCO Object Detection and Instance Segmentation

To address the concern that the experiment in the paper is not enough, we assess the performance of AtomNAS models as feature extractors for object detection and instance segmentation on COCO dataset. For more details, please check the appendix A at the end of our revised paper.

We first pretrain AtomNAS models (without Swish activation function and Squeeze-and-Excitation (SE) module) on ImageNet, use them as drop-in replacements for the backbone in the Mask-RCNN model, by building the detection head on the last feature map, and finetune the model on COCO dataset.

We use the open-source code MMDetection (https://github.com/open-mmlab/mmdetection ). All the models are trained on COCO train2017 with batch size 16 and evaluated on COCO val2017. Following the schedule used in the open-source implementation of TPU-trained Mask-RCNN (https://github.com/tensorflow/tpu/tree/master/models/official/mask_rcnn ), the learning rate starts at 0.02 and decreases by a scale of 10 at 15-th and 20th epoch respectively. The models are trained for 23 epochs in total.

The results are shown below. The detection results of baseline models are from [1]. We can see that all three AtomNAS models outperform the baselines on both object detection task. The results demonstrate that our models have better transferability than the baselines.

| Model                           | FLOPs | Cls (%)   | detect-mAP (%)   | seg-mAP (%)  |
| ------------------------- | -------- | ----------- | ---------------------- | ------------------ |
| MobileNetV2              | 301M  | 73.6         | 30.5                          | -                            |
| Proxyless (mobile)  | 320M  | 74.6         | 32.9                          | -                            |
| SinglePath+               | -            | 75.6         | 33.0                          | -                            |
| AtomNAS-A                | 258M  | 74.6         | 32.7                          | 30.1                     |
| AtomNAS-B                | 326M  | 75.5         | 33.6                          | 30.8                     |
| AtomNAS-C                | 360M  | 75.9         | 34.1                          | 31.4                     |
| ------------------------- | -------- | ----------- | ---------------------- | ------------------ |


=== References ===
[1] Stamoulis, et al., Single-Path Mobile AutoML: Efficient ConvNet Design and NAS Hyperparameter Optimization. In CoRR, 2019.

---

### Decision · Program_Chairs · 2019-12-19

**Decision:**

Accept (Poster)

**Comment:**

Reviewer #1 noted that he wishes to change his review to weak accept post rebuttal, but did not change his score in the system.  Presuming his score is weak accept, then all reviewers are unanimous for acceptance.  I have reviewed the paper and find the results appear to be clear, but the magnitude of the improvement is modest.  I concur with the weak accept recommendation.